# Artificial intelligence algorithms predict the efficacy of analgesic cocktails prescribed after orthopedic surgery

**Gerhard Fritsch**[1,2]*, **Heinz Steltzer**[3,4], **Daniel Oberladstaetter**[1], **Carolina Zeller**[4], **Hermann Prossinger**[5]

**1** Department of Anesthesiology and Intensive Care Medicine, AUVA Trauma Hospital Salzburg, Salzburg, Austria, **2** Paracelsus Medical University, Salzburg, Austria, **3** Department of Anesthesiology and Intensive Care Medicine, AUVA Trauma Center Vienna, Meidling, Austria, **4** Sigmund Freud University Vienna, Austria, **5** Department of Evolutionary Biology, University of Vienna, Vienna, Austria

* gerhard.fritsch@auva.at

## Abstract

### Background

Mixtures ('cocktails') of various analgesics are more effective in controlling post-operative pain because of potential synergetic effects. Few studies have investigated such effects in large combinations of analgesics and no studies have determined the probabilities of effectiveness.

### Methods

We used one-hot encoding of the categorical variables reported pain levels and the administered cocktails (from a total of eight analgesics) and then applied an unsupervised neural network and then the unsupervised DBSCAN algorithm to detect clusters of cocktails. We used Bayesian statistics to classify the effectiveness of these cocktails.

### Results

Of the 61 different cocktails administered to 750 patients, we found that four combinations of three to four analgesics were by far the most effective. All these cocktails contained Metamizole and Paracetamol; three contained Hydromorphone and two contained Diclofenac and one Diclofenac-Orphenadrine. The ML probability that these cocktails decreased pain levels ranged from 0.965 to 0.981. Choice of a most effective cocktail involves choosing the optimum in a 4-dimensional parameter space: maximum probability of efficacy, confidence interval about maximum probability, fraction of patients with increase in pain levels, relative number of patients with successful pain level decrease.

### Conclusions

We observed that administering one analgesic or at most two is not effective. We found no statistical indicators that interactions between analgesics in the most effective cocktails decreased their effectiveness. Pairs of most effective cocktails differed by the addition of

**Data Availability Statement:** All relevant data are within the paper and its Supporting Information files.

**Funding:** This scientific work was supported by the Austrian Working Insurance Company (AUVA), Wienerbergstrasse 11, 1100 Wien, Austria.

**Competing interests:** There exist no competing interest from any party involved in this project.

only one analgesic (Diclofenac-Orphenadrine for one pair and Hydromorphone for the other). We conclude that the listed cocktails are to be recommended.

## Introduction

The alleviation of postoperative pain by administering analgesics after orthopedic surgery is a major issue in perioperative medicine. Especially in cases of shoulder surgery (e.g. rotator cuff repair), total joint replacement and extremity trauma, pain levels are expected to be high; it is therefore imperative to apply highly effective strategies with the aim to enhance recovery, avoid patient discomfort and suffering, and reduce the risk of pain-related complications [1–3].

In multimodal pain therapy, physicians usually administer combinations of two or more analgesics [4, 5]. Due to possible pharmacological interactions among these, the prediction of efficacy is rarely known. Interactions can be based on mechanisms of action (e.g. pharmacodynamics on receptors) or pharmacokinetic pathways. For instance, up to 95% of Diclofenac is bound to serum albumin after absorption and renal elimination takes place after hydroxylation by CYP 3A4 and glucuronidation in the liver. Clinical effects are achieved by the blockade of cyclooxygenase I and II, resulting in decreased synthesis of plostaglandins. Acetaminophen also exhibits its action through the cyclooxygenase pathway in inhibiting prostaglandin synthesis. Opioids, on the other hand, act via receptors, which are specific for these analgesics and they are eliminated after hydroxylation by CYP 3A4 in the liver. Thus the combination of more than 2 drugs can be unclear and confusing for the clinician regarding the net-effect of prescriptions.

Postoperative prescriptions contain opioids and non-opioids with various pharmacokinetic and pharmacodynamic properties. To our knowledge, data about how these properties may change when administered in combinations of more than two is sparse and rarely, if ever, published [6–11]. A serious challenge to any statistical evaluation of analgesic medications with large numbers of combinations of analgesics arises: drawing conclusions using conventional statistical methods is fiendishly difficult. We use artificial intelligence methods [12–15] to overcome this difficulty.

We use (artificial) neural networks (NNs) for our data analysis; specifically, we use the unsupervised neural networks (Fig 1) called autoencoders. These unsupervised neural networks generate output that mimics the input to high precision (hence their name: autoencoders) by minimizing the loss (the sum of squares of the differences between input and output, averaged over the training set). We then use the weightings of the code layer for each input feature vector as the coordinates of the dimension-reduced feature vector (Fig 1). Feature vectors are described in the next paragraph.

The pain levels are categorical variables, as are the analgesics administered. We use one-hot encoding to generate a 38-dimensional feature vector for each patient (see Methods section). These feature vectors are not independent. A dimension reduction algorithm (a neural-network autoencoder) finds independencies and maps the result onto a two-dimensional manifold, a planar map. Each patient is a point in this plane, and the points are not randomly distributed; rather, they are clustered. Of the many clustering algorithms at our disposal, we use the DBSCAN clustering algorithm [16], because applying it to the points identifies clusters of cocktails with many analgesics in common. The interdependencies generate clusters containing highly effective analgesics; a discovery which cannot, as we discuss below, be found in any other way (there are 61 different cocktails of analgesics and a total of 750×2 = 1500 pain

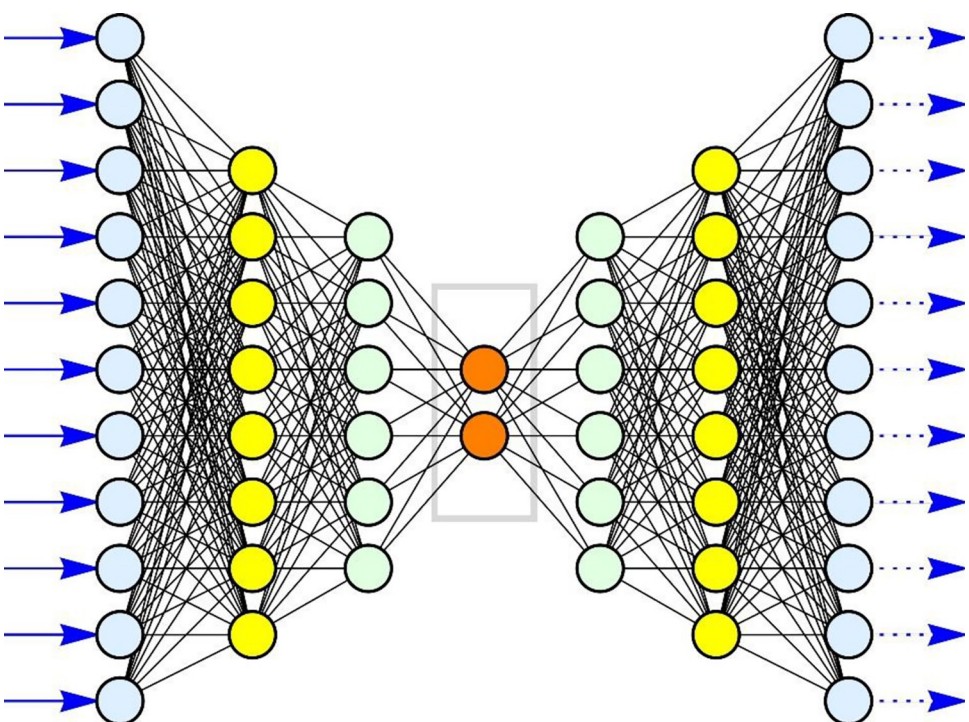

**Fig 1. A symbolic rendition of an autoencoder.** The 11 inputs (in the drawing) are represented by blue arrows from left to right. The inputs are conventionally labelled neurons (hence the name "neural network"). Each input neuron (light blue) has as many outputs as there are neurons in the next layer (8 in the drawing, represented as yellow discs). Each 'yellow' neuron thus has 11 inputs (represented as thin black lines, called edges). Each neuron in the yellow layer has as many outputs as there are neurons in the next layer: 6 outputs for each 'yellow' neuron and therefore 8 inputs for each 'green' neuron of the next layer. And so it continues: each 'green' neuron has as many outputs as there are neurons in the next layer (consisting of two 'orange' neurons) and each 'orange' neuron has 6 inputs. The light blue neurons on the left are called the input layer, the light blue neurons on the right are called the output layer. The (two) yellow layers, the (two) green layers and the (one) orange layer are called the hidden layers. An autoencoder always has the same number of output neurons as it has input neurons. The number of hidden layers is part of the design by the engineer constructing the autoencoder, as well as the number of neurons in each hidden layer. The numerical values along the black edges between neurons are determined by an algorithm. The autoencoder attempts to produce an output equal to the input (hence the name 'autoencoder') without being an identity mapping. An important feature for modern autoencoders is the ability to cut (set to zero) certain interconnections (edges), or make them numerically very small (usually by using a sigmoid function). The central layer is called the code. If the inputs are the feature vectors, then the numerical values of the code are the components of the dimension-reduced feature vector. In the drawing: the input feature vector has 11 dimensions and the dimension-reduced feature vector has 2 components. Mathematically: if this is a successful autoencoder, it has detected linear combinations between the components of the (input) feature vector that can be represented by two variables in a 2D space.

level registrations). We have attached a GLOSSARY (S1 File) containing a more detailed description.

The data set in this study consists of categorical (nominal) variables. Multinomial logistic regression is therefore not possible, unless one makes two assumptions: one, it is possible to distinguish between explanatory and dependent variables, and two, it is possible to map the categories of the explanatory variables into cardinal integers. Both these assumptions are violated [17], so we need not enter the discussion of how to deal with the issue of the 'curse of dimensionality' [18]. Because the relations between the various categorical variables cannot be assumed to be linear, nonlinear mappings, if found, will have too many parameters [19]. Autoencoders deal with all the above problems automatically: they incorporate nonlinear mappings, dimension-reduce the input feature vectors, and also operate on categorical variables, because these have been transformed into feature vectors via one-hot encoding [13].

## Methods

The study was performed after approval by the local ethics committee (Ethikkommission der Krankenanstalten der AUVA; No.13/2020, July 1[st], 2020). The need for informed consent was waved by the ethics committee due to the retrospective character of the study. Data were fully anonymized before the analyze.

The data was collected prospectively in conjunction with the peri- and postoperative pain-visits between March 2018 and December 2019 in one single emergency hospital and was analyzed retrospectively. Patients underwent various orthopedic procedures (Table 1) during the routine operating-room-schedule between 8 am and 4 pm. Inclusion Criteria were: shoulder surgery, hip- or knee replacement surgery, repair of the anterior cruciate ligament, spine surgery, amputation of limb and complex reconstructive procedures. Patients were excluded only if the data was incomplete. Diclofenac (2 × 75 mg within 24 hours), Metamizole (3 × 1000 mg within 24 hours), Diclofenac/Orphenadrine (2 × 75 mg / 2 × 30 mg within 24 hours), and Paracetamol (3 × 1000 mg within 24 hours) were administered intravenously according to a fixed dose-plan. Hydromorphone (2 × 2 mg modified release—or 2 × 4 mg modified release for patients with body mass/weight > 80 kg within 24 hours), Tramadol (2 × 150 mg within 24 hours) and Dexibuprofen (2 × 400 mg within 24 hours) were prescribed orally; Piritramide (75 mg; only on demand) was administered subcutaneously. Hydromorphone (1.3 mg or 2.6 mg, depending on body mass/weight) was prescribed as rescue medication for patients who had also received the modified dose of Hydromorphone. Date of birth, date of operation and biological sex (not gender) of each patient was also recorded, as well as the mode of anesthesia.

**Table 1. Orthopedic procedures.**

| Orthopedic procedures | Number of Cases |
|---|---:|
| Surgery of peripheral nerves | 14 |
| Spine surgery | 20 |
| Shoulder surgery | 179 |
| Bone surgery upper limb | 118 |
| Bone surgery hand | 21 |
| Shoulder arthroplasty | 20 |
| Surgery elbow | 37 |
| Soft tissue surgery upper extremity | 10 |
| Femur neck | 55 |
| Calf bone surgery | 42 |
| Bone surgery foot | 20 |
| Hip arthroplasty | 37 |
| Knee surgery | 315 |
| Knee arthroplasty | 32 |
| Ankle surgery | 13 |
| Soft tissue surgery calf | 37 |
| Metal removal | 35 |
| Amputation lower extremity | 7 |
| Soft tissue surgery (other) | 92 |
| **Total** | **1104** |

The listing of the orthopedic procedures performed during the study period (March 2018–December 2019). The number of orthopedic procedures is larger than the number of patients, because some patients had more than one orthopedic procedure.

Maximum pain levels during movement directly after operation and minimum pain levels during physicians' visits as felt by the patients were documented. This part of the data set thus consists of the lists of which combinations of 8 analgesics (Table 3) were administered to 750 patients (421 males and 329 females; 52% general anesthesia, 48% regional anesthesia) in this hospital. Patients were asked as to their perceived pain levels on the 11-part ordinal numeric rating scale (NRS) as *categorical* entries from 0 ("A") to 10 ("K") in both of the aforementioned cases: (a) maximum pain during motion/flexing prior to the pain visit, and (b) pain at rest at the time of the pain visit between 8 am and 2 pm on postoperative Day 1.

If the ages of patients are normally distributed, then 'age' would be a confounding variable; we must test this possibility. The ages of the patients were calculated to the nearest day using a Gregorian calendar algorithm that outputs the number of days between the date of birth and the date of the operation (this method avoids statistical uncertainties due to rounding errors that occur when ages are given in years). Conventional descriptors and point estimators are listed in Table 2. The ages were converted to fractions of years and then an AI algorithm, KDE (kernel density estimation) [13], was used to find the likelihood distribution (*pdf*) of the ages. Mode (age at maximum likelihood) and expectation value ($E = \int_0^1 x \, pdf(x) \, dx$) of this distribution were calculated, as well as the uncertainty interval $HDI_{95\%}$ (highest density interval at 95% confidence [20]; further details in Glossary). This interval is defined as having endpoints with equal likelihood and 95% probability of age; it is the uncertainty measure that is necessary when the distribution is not a parametric one [20]. The resulting *pdf* shows that ages of patients are very close to being uniformly distributed, so age need not be considered a confounding variable.

We use one-hot encoding [13] to convert the categorical variables (both pain levels and identifiers of analgesics) to feature vectors. Converting pain levels encoded as ordinal numbers to cardinal numbers does *not* lead to proper statistical analyses [17]. (Ordinal numbers are used as indices. Thus, the intervals between ordinal pain levels are not numerically defined. "It should be pointed out that numbers may be arbitrarily associated with each category, but this fact in no way justifies the use of the usual arithmetic operations on these numbers." [17]) For example: if a patient registers pain levels "H", and "B" at the two times specified above, then the pain level vectors are $(0,0,0,0,0,0,0,1,0,0,0)^T$, and $(0,1,0,0,0,0,0,0,0,0,0)^T$ respectively. For each analgesic, the encoding vector is either $(1,0)^T$ (if not administered) or $(0,1)^T$ (if administered). In the analysis presented in this paper, pain level shifts from the upper extreme

**Table 2. Patients' age.**

| Descriptor | Age (years) |
|---|---|
| Range | 7.9–97.8 |
| Mode | 52.1 |
| Expectation | 47.4 |
| $HDI_{95\%}$ | 14.5–81.0 |
| Arithmetic mean | 47.3 |

The descriptors of the ages of the patients. Ages (in days) were calculated as the difference between date of birth and date of operation/surgery, using a Gregorian calendar algorithm and then converted to years. The distribution of ages has been estimated using kernel density estimation (KDE) and is far from normal; indeed, it is far from any parametric (continuous) distribution. We note that (1) the mode and the expectation value are far apart, indicating that the age distribution is not symmetric; (2) the arithmetic mean is not a good estimator of the expectation value, despite 750 data points (3) the confidence interval $HDI_{95\%}$ (details in the Glossary and in reference [20]) shows that at 95% confidence, very few ages are not included in the KDE distribution.

**Table 3. Analgesics and their distribution in clusters.**

| Analgesic | Abbreviation | Number of patients with administered analgesic |
|---|---|---|
| Piritramide | Pir | 21 |
| Hydromorphone | Hyd | 547 |
| Tramadol | Tra | 17 |
| Dexibuprofen | Dex | 6 |
| Diclofenac-Orphenadrine | DiO | 396 |
| Metamizole | Met | 647 |
| Diclofenac | Dic | 264 |
| Paracetamol (Acetaminophen) | Par | 694 |

| Cluster | Number of Patients | Number of Registrations | Number of Cocktails |
|---|---|---|---|
| 1 | 49 | 31 | 2 |
| 2 | 449 | 241 | 20 |
| 3 | 54 | 34 | 5 |
| 4 | 38 | 37 | 21 |
| 5 | 128 | 88 | 13 |
| 6 | 32 | 21 | 1 |

| Cluster | Cocktail | Number of Patients | Fraction in Cluster (%) | Fraction Overall (%) |
|---|---|---|---|---|
| 1 | Met-DiO-Hyd | 39 | 80 | 5 |
| 2 | Met-Par | 40 | 9 | 5 |
| | Hyd-Met-Par | 88 | 20 | 12 |
| | DiO-Met-Par | 39 | 9 | 5 |
| | Hyd-DiO-Met-Par | 164 | 37 | 22 |
| 3 | Hyd-Dic-Par | 49 | 91 | 7 |
| 4 | various (22) | 38 | - | 5.1 |
| 5 | Met-Dic-Par | 47 | 37 | 6 |
| | Hyd-Met-Dic-Par | 42 | 33 | 6 |
| 6 | Dic-Par | 32 | 100 | 4 |

*Top panel*: A listing of all eight analgesics administered at least once to at least one patient. Because several patients received cocktails of more than one analgesic, the right column total exceeds 750, the number of patients. *Central panel*: The distribution of the 750 patients, 452 registrations, and 61 cocktails in the six clusters obtained by dimension-reducing the feature vectors onto a 2-dimensional manifold using an autoencoder and clustering the registrations using the DBSCAN algorithm. The cocktails are graphed as 241 points in **Fig 2**. *Lower panel*: The most occurring cocktails (i.e. administered to above 4% of the 750 patients), as distributed by cluster. None of these cocktails consists of a single analgesic. Cluster #4 consists of 22 cocktails, but none have a fraction above 1%. Only Cluster #1 has a cocktail without Par. Cluster #2 always has Met and Par in its most-occurring cocktails. Cluster #5 always has Met-Dic-Par triple in its most-occurring cocktails. The cocktail Dic-Par forms its own cluster.

(maximum directly after operation during movement) to the lower extreme (minimum at visit while at rest), may depend on which cocktails were administered. The (categorical) registration vectors for *each* analgesic and for *each* pain level are individually orthonormal, ensuring independency and ensuring no pre-defined metric. For each patient, the feature vector resulting from concatenation of the registration vectors has dimension 2×8+11+11 = 38. Because the 38-dimensional feature vectors of the 750 patients are not orthogonal, interdependencies exist. We find these by using an unsupervised artificial intelligence algorithm involving a neural network [13, 14]; more specifically, we use an autoencoder with seven layers in order to detect the

registrations of each patient on a 2-dimensional manifold. We then apply the DBSCAN clustering algorithm [16] to find clustering patterns in the outcomes. Finally, we investigate how registrations are distributed within these clusters.

Because there is an inherent uncertainty as to whether an administered cocktail is effective, we need a Bayesian approach to determine not only this uncertainty, but also the ML probability, which is the mode of the probability (in Bayesian statistics, probability is a random variable) that a shift to lower pain levels is to be expected. The maximum likelihood of shifts to lower pain levels is determined by first inventorying the number of decreases ($n_1$) and the number of increases or zero shift ($n_2$) for each analgesic cocktail and then calculating the mode of the Beta function Bε($n_1$+1, $n_2$+1). The mode informs us of the ML probability of finding a shift to lower pains levels. The uncertainty of the mode—the HDI$_{95\%}$ interval [20]—is determined by the area under the likelihood function that has the same likelihoods at the interval ends and area = 0.95. If the HDI$_{95\%}$ interval includes $s = \frac{1}{2}$, then the shifts to lower pain levels for the specified cocktail are not significant at a 95% confidence interval.

## Results

The number of possible cocktails for 8 analgesics is $\begin{Bmatrix} 8 \\ 2 \end{Bmatrix} = 127$, where $\begin{Bmatrix} n \\ k \end{Bmatrix}$ denotes a Stirling number of the second kind [21]. Of these 127 possible cocktails, only 61 were administered (Table 2). A list of all 61 cocktails (including the inventories of their pain level shifts) is too large for inclusion in this manuscript.

At visit without movement, only the pain levels A–J were communicated by the patients. We observed that the 2-dimensional registrations cluster in only six clusters (Fig 2). These six clusters are characterized by specific cocktails (Fig 3 shows a part of the cocktail spectra in Cluster #2) that have some analgesics in common. Not all medications occurred in each cluster, and no cocktail occurred in more than one cluster. We also observed that rarely occurring cocktails (i.e. cocktails administered to at most 4 patients) occurred predominantly in only one cluster (Cluster #4; Table 3); indeed, this one cluster contained *only* rarely occurring cocktails. Table 3 also lists the distribution of the most frequently occurring cocktails occurring in Clusters #1–#3 and #5–#6.

Every cocktail in every cluster showed a decrease in pain levels, albeit of different magnitude for different cocktails and with differing frequencies. In each cluster, only one or two cocktails were administered most frequently (not shown).

## Discussion

In our investigation we could show the efficacy of different combinations of postoperative pain medications. A combination of 4 drugs revealed to be the most effective. It consisted of Hydromorphone, Metamizole, Paracetamol and Diclofenac/Orphendrine. We observed that the cocktail Hyd-DiO-Met-Par outperforms the others, although Met-Dic-Par is a close contender. Hyd-DiO-Met-Par has the highest probability (0.981) of effectively decreasing the pain levels, with a mode of 34 shifts at 4 pain levels *downward*. The fraction of patients that report an increase in pain level shifts is small (1.8%), much lower than all the others, except for Met-Dic-Par. We observe that DiO-Met-Par also exhibits a very high performance, comparable to Hyd-DiO-Met-Par; the former lacks Hyd. We conclude that the statistical analysis infers that adding Hyd to the cocktail DiO-Met-Par will increase its performance, albeit not very much. This outcome can be explained by the strength of Hydromorphone, the only opioid in this cocktail. In former studies combinations of analgesics have already been proven to be effective

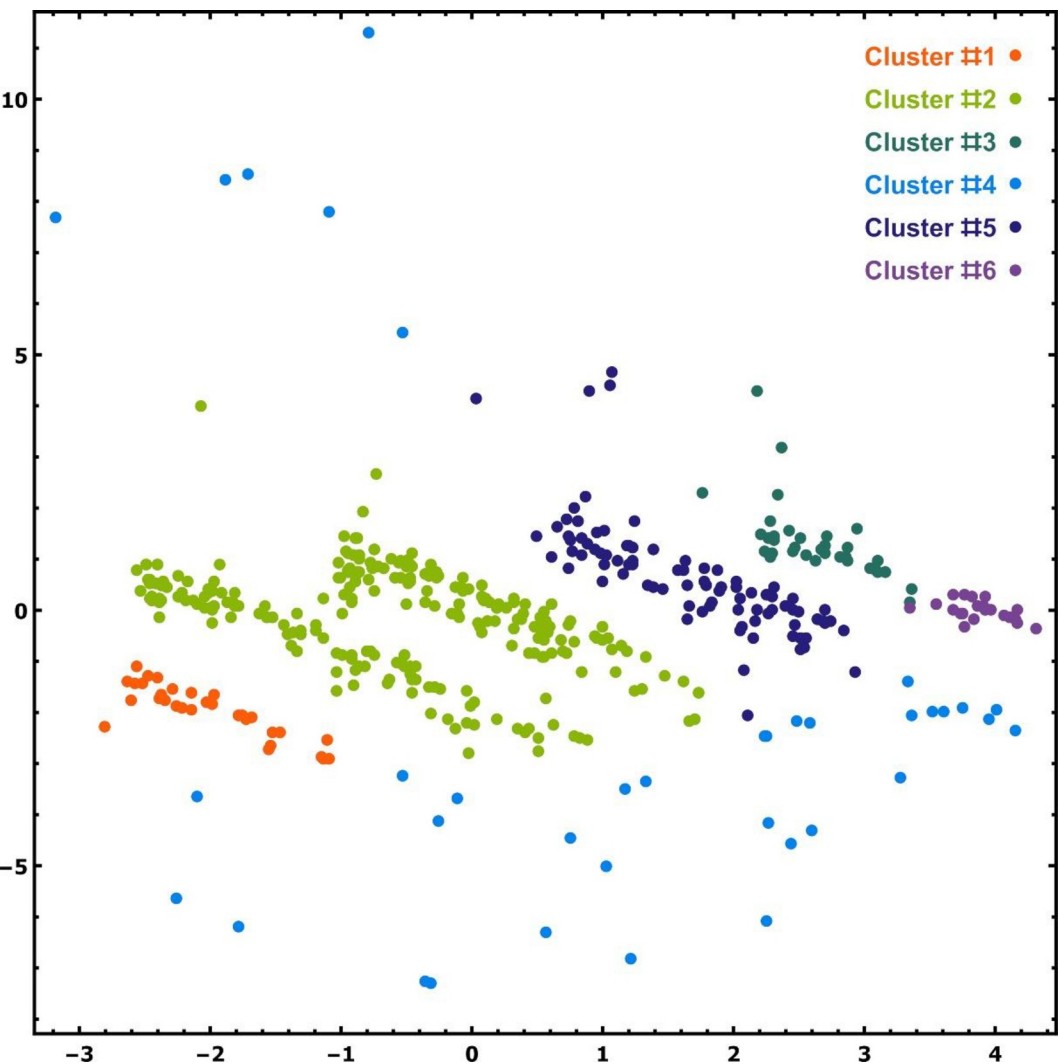

**Fig 2. The 452 registrations of the 750 feature vectors after dimension reduction using the AI algorithm autoencoder, the 6 clusters obtained with the DBSCAN clustering algorithm are color-coded.** Not all clusters are localized (there are points of Cluster #4 above and below the other clusters). Some points have a high multiplicity (Table 2); therefore, some cluster members may be rendered by one point, which may be the registrations of many patients. The $x$- and $y$-coordinates are for calculation and statistical analysis purposes only; they have no interpretable, holistic meaning.

[6–11]. Only the number of drugs in combination has been much lower than in our study. The maximum number was 3, whereas in our investigation we studied a total of 8 different prescriptions.

Analyzing the effects of various cocktails on pain level decrease or increase (if observed) is a multivariate problem. Furthermore, because the random variables pain level and cocktail label are categorical, parametric distributions of continuous random variables are impossible—therefore point estimators, such as means and standard deviations, are undefined. If we were to compare the distributions of the 11 pain levels for each of the 61 cocktails (within each cocktail, the pain level distribution is a Dirichlet distribution), we would have to pore over 61 multivariate distributions, few of which have the same number of random variables. The output would have to be in a stacked/layered table form; and, unfortunately, we humans cannot envision *pdf*s of distributions in more than 3D. The exercise of poring over 61 Dirichlet

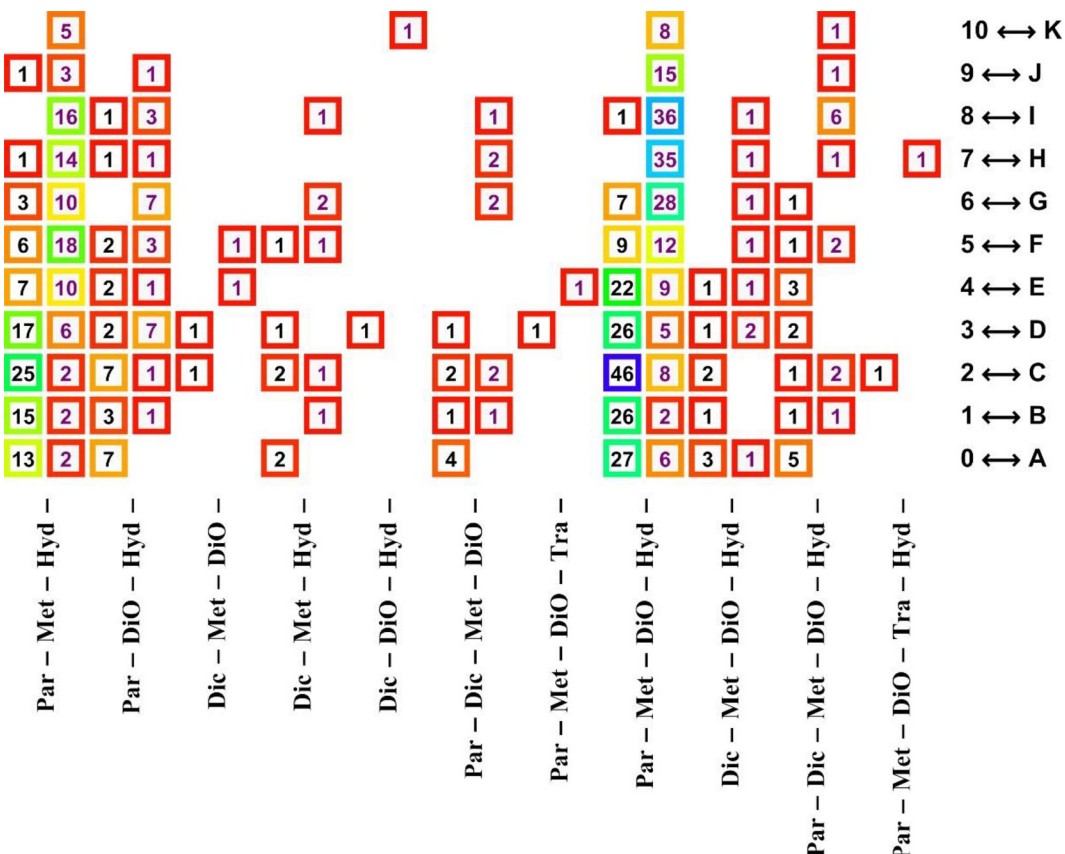

**Fig 3. A selection of pain distributions reported by patients in Cluster #2.** For each cocktail there are two columns. In each column, the number of patients registering their pain levels (left: motionless during visit, right: maximum while flexing shortly after operation) are written in squares with color-coded borders (red: lowest; purple: highest). The border color encodes the fraction of pain registrations for the administered cocktail. For example, 36 patients registered pain level "H" (ordinal number "7") after operation when administered the Par-Met-DiO-Hyd cocktail. At visit, the minimum pain was pain level "C" (ordinal number "2") for 46 patients. The color coding shows the dramatic decrease in pain levels for the cocktails Par-Met-Hyd and Par-Met-DiO-Hyd (roughly "H"→"B" for both these cocktails).

distributions is not only very tiring, but also error-prone. Consequently, we let a neural network analyze the data for us, as it is intelligent and far superior to human analysis skills for the sheer number of possible outcomes. In the AI approach we implement, namely using an (unsupervised) autoencoder and the DBSCAN clustering algorithm, we discover clusters, pain shifts within each cluster, and the distributions of the administered cocktails—all "in one go."

The feature vectors of patients have internal dependencies, which the autoencoder discovered (and we could not). We do not classify outcomes (how the pain level shifts related to cocktails administered) by training the neural network on a training set of known outcomes; this latter approach would be supervised learning. On the contrary, we did not know any of the outcomes beforehand; they were all discovered by the neural network. Furthermore, we used the (highly effective) clustering algorithm DBSCAN, which finds clusters (and how many) in an unsupervised manner. We find that the combination of the autoencoder and the DBSCAN algorithm clustered cocktails together with common analgesics as ingredients in cocktail clusters. This is very surprising result was not foreseen, but discovered by the statistical analysis.

Cocktails with certain combinations of analgesics cluster. The (perhaps surprising) insight: in all clusters (except Cluster #4, which contains the quasi-noise signal), the cocktails with a

strong signal consist of a base combination of a few of the eight analgesics plus perhaps one or some other further one. We therefore address the issue of whether the addition of an analgesic to the base combination is (medically) indeed necessary. For example, in Cluster #2, a base cocktail is Hyd-Met-Par, and another cocktail in the same cluster contains Hyd-DiO-Met-Par. We therefore ask whether this addition of DiO to the cocktail is medically necessary. The cocktail Hyd-DiO-Met-Par is not more effective than Hyd-Met-Par (in a statistical sense): the pain shift (decrease) is statistically just as large. We note that discovering this particular cocktail pair (and the attendant comparability of pain level shifts) is impossible without the implementation of neural networks.

Clinicians might expect this phenomenon (presumably the reason why some in this hospital administered cocktails). The concept of combining non-opioids with opioids in the application of multimodal postoperative pain-therapy is thus confirmed by our analysis [4, 10]. When comparing different high-performing cocktails, we need to consider four parameters together (thus the evaluation is a four-dimensional problem): (1) the mode of the probability of effectiveness, (2) the $HDI_{95\%}$ uncertainty, (3) the fraction of patients that experience an increase in pain level shifts, and (4) the mode of the pain level shifts. Given the combinations of these four parameters, the cocktail Hyd-DiO-Met-Par outperforms all others.

We had no control over which cocktails were administered; this study is not a controlled trial because it included a large number of different surgical procedures in a hetrogenious patient-sample avoiding the use of a conventional study protocol (this study allows neural networks to show their full analytic strength, thereby possibly avoiding placebo-type trials). Trained neural networks produce output by finding non-linear weightings of many inputs; thus avoiding fragmentation of studies that include fragmentation due to many confounding factors and the curse of dimensionality. In the hospital, certain cocktails are preferred by some clinicians, while others have other preferences. The number of different cocktails is larger than the number of clinicians administering these, so we cannot uniquely map cocktails with clinicians (nor would the Ethics Committee allow us to do so). Again: unsupervised neural networks that dimension-reduce the data and clusters outputted by DBSCAN shows the power of our approach over controlled studies involving one or two (rarely three) cocktails and perhaps a placebo group. The data set containing a large number (61) of different cocktails ensures that our analysis outcome is not skewed due to the lack of some cocktails not being administered. In fact, of the 127 possible cocktails, most cocktails *not* administered were rare pairs (21 pairs —~34%—of 61 cocktails were in Cluster #4). We also note that 61 of 127−8 = 115 cocktails were administered (*far better* than $\frac{61}{115} \approx 54\%$, because no cocktails with more than four analgesics were administered), so we argue that cocktail diversity and effectiveness was more than reasonably monitored.

Pain medications in our study were not fixed in the study protocol; ***this*** would be a limitation when data is analyzed with frequency tables. Although the study-sample size was large, some effects of cocktails could perhaps not be discovered as their signal could have been below the noise level in the analysis of this data set. This would only be a limitation if one were to assume that *every* small effect has an ***explanation***—but noise effects do not. Given the power of neural networks of detecting small effects, the hypothesis that what we observed as noise is actually an effect that cannot be detected due to small sample size must be considered a far-fetched speculation. Because no clinicians in this hospital administered cocktails with more than four analgesics, a limitation in our conclusion ***might*** be the implication that cocktails with more than four analgesics can be more effective—an objection we also consider borderline, because of the extremely high efficacy of many of the cocktails with three or four analgesics (Table 4).

**Table 4. Pain level shift and analgesic cocktails.**

| Cluster | Cocktail | $mode_{shift}$ | $HDI_{95\%}$ | % | Pain Level Shift Distributions | | | | | | | | | | | | | | |
|---|---|---|---|---|---|---|---|---|---|---|---|---|---|---|---|---|---|---|---|
| | | | | | −4Δ | −3Δ | −2Δ | −1Δ | 0Δ | +1Δ | +2Δ | +3Δ | +4Δ | +5Δ | +6Δ | +7Δ | +8Δ | +9Δ | +10Δ |
| 1 | Met-DiO-Hyd | 0.973 | 0.881–0.999 | 2.6 | | 1 | | | 2 | 2 | 4 | 5 | 12 | 8 | 3 | 2 | | | |
| 2 | Met-Par | 0.994 | 0.836–0.991 | 5.0 | | | 1 | 1 | 4 | 6 | 8 | 4 | 7 | 4 | 3 | 2 | | | |
| | Hyd-Met-Par | 0.965 | 0.910–0.991 | 3.4 | | 1 | 1 | 1 | 3 | 9 | 10 | 19 | 22 | 8 | 3 | 6 | 4 | 1 | |
| | DiO-Met-Par | 1.000 | 0.926–1.000 | 0 | | | | | 1 | 2 | 6 | 9 | 12 | 4 | 3 | 2 | | | |
| | Hyd-DiO-Met-Par | 0.981 | 0.950–0.995 | 1.8 | 1 | 1 | 1 | | 7 | 17 | 11 | 21 | 34 | 24 | 26 | 11 | 7 | 2 | 1 |
| 3 | Hyd-Dic-Par | 0.933 | 0.835–0.983 | 6.1 | 1 | 1 | | 1 | 4 | 3 | 6 | 13 | 8 | 7 | 2 | 2 | 1 | | |
| 4 | various | - | - | - | | | | | | | | | | | | | | | |
| 5 | Met-Dic-Par | 0.977 | 0.898–0.999 | 2.1 | | 1 | | | 3 | 7 | 6 | 7 | 10 | 7 | 2 | 1 | 1 | 2 | |
| | Hyd-Met-Dic-Par | 0.976 | 0.894–0.999 | 2.4 | | 1 | | | | 4 | 5 | 8 | 6 | 4 | 8 | 4 | 2 | | |
| 6 | Dic-Par | 0.933 | 0.807–0.989 | 6.2 | | | 2 | | 2 | 4 | 4 | 7 | 4 | 5 | | 2 | | 1 | 1 |

Likelihoods of decrease in pain level shifts (Δ) and their distributions, categorized by clusters. The shifts are *not* numerical values, because the pain levels are categorical variables. The column labelled $mode_{shift}$ lists the ML probability of observing a pain level shift that is *not an increase* in reported pain levels; the column labelled $HDI_{95\%}$ shows the uncertainty interval of this ML probability (details in the Methods section). We observe that all ML probabilities are very close to 1.00..., so we conclude that the cocktails listed here are very effective. % is the fraction of patients for this cocktail that reported an *increase* in pain levels between maximum at motion and minimum at visit despite being administered the cocktail listed in this row. The numbers are the number of patients reporting the pain level shift specified in the rectangle. We note that a few patients report an increase in pain levels (negative pain level shifts), and several no pain level shift.

## Conclusion

We analyzed the effectiveness of 61 analgesic cocktails administered to 750 patients undergoing many different orthopedic surgical procedures using several unsupervised artificial intelligence algorithms. The cocktails were mixtures of two to four analgesics and we found that the three most effective cocktails were (from highest to lowest): (a) Hyd-DiO-Met-Par, (b) Met-Dic-Par, and (c) DiO-Met-Par. For the clinician (but not for the patient), medication/cocktail choice is a four-dimensional challenge: (1) The ML probability (the mode) of lowering pain levels; (2) The uncertainty (the CI) of the ML probability of pain level shift to lower pain levels; (3) The mode of the frequency of patients reporting the large pain level shifts to lower pain levels; (4) The small fraction of patients not experiencing a pain level shift to lower pain levels. For the clinician, the importance of the four criteria listed above are not to be underestimated, nor should any one of them be neglected when making a therapeutic decision. For the patient, on the other hand, the ML probability and its CI is of marginal importance.

We do not supply a prescriptive regimen for all or many clinical situations. We do, however, present a method that shows that many confounders may have no effect when choosing an optimal pain reduction therapy.

## Supporting information

**S1 Fig. An example of the descriptors of an unsymmetrical distribution.** For ease and clarity of description, we show the *pdf* (the likelihood function Λ(s)) of a Beta distribution, namely that of a Bayesian likelihood. The maximum likelihood is the mode, and the expectation is . The confidence interval HDI$_{95\%}$ is shown via a double-ended arrow. The likelihoods at the ends ($s_1$ and $s_2$) of the confidence interval are equal: Λ($s_1$) = Λ($s_2$). The shaded area is 95%. (TIF)

**S1 Data.**
(XLSX)

**S1 File. Glossary.**
(DOCX)

## Acknowledgments

We thank Ernst Reitbichler and the anesthesia nursing-team at the TZW Lorenz Boehler for their contribution in collecting data and Christopher Lockie, MD for language editing.

## Author Contributions

**Conceptualization:** Gerhard Fritsch, Heinz Steltzer, Daniel Oberladstaetter, Carolina Zeller, Hermann Prossinger.

**Data curation:** Daniel Oberladstaetter, Carolina Zeller, Hermann Prossinger.

**Formal analysis:** Carolina Zeller, Hermann Prossinger.

**Funding acquisition:** Gerhard Fritsch.

**Investigation:** Gerhard Fritsch, Hermann Prossinger.

**Methodology:** Gerhard Fritsch, Hermann Prossinger.

**Project administration:** Gerhard Fritsch, Carolina Zeller.

**Writing – original draft:** Gerhard Fritsch, Heinz Steltzer, Hermann Prossinger.

**Writing – review & editing:** Heinz Steltzer, Daniel Oberladstaetter, Carolina Zeller, Hermann Prossinger.

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
