## [Decision Letter · Decision Letter 0]

4 Feb 2022

PONE-D-21-40117Artificial intelligence algorithms predict the efficacy of analgesic cocktails prescribed after orthopedic surgeryPLOS ONE

Dear Dr. Fritsch,

Thank you for submitting your manuscript to PLOS ONE. After careful consideration, we feel that it has merit but does not fully meet PLOS ONE’s publication criteria as it currently stands. Therefore, we invite you to submit a revised version of the manuscript that addresses the points raised during the review process.

Please, revise methods section to clarify clinical variables  and discussion to contextualize the clinical limitations of this paper as requested by the reviewer 1. 

Specific comments:

1. line 21 consider "potential" instead of "claimed" - efficacious synergy has been established.

2. line 46 - be specific about certain aspects of orthopaedic surgery - particularly the interest in "outpatient" total joints in the US, extremity trauma, and shoulder surgery (e.g. rotator cuff repair) are the most painful orthopedic surgeries. This is where pain control is most needed.

3. line 50 - This sentence requires a little more explanation. What specific efficacy are you speaking about?

4. line 57 - It would be useful to in 1 or 2 sentences explain why conventional logistic regression cannot answer this question. Those unfamiliar with ML will not understand this.

5. line 59 - the first part of this sentence is confusing. Specifically the part "to learning algorithms labelled AI (artificial intelligence); the latter". You could get rid of that and the sentence is less confusing to read for the novice. ML really comes in 3 types - supervised, unsupervised, and reinforcement learning. In general lines 59-67 don't add much to this introduction. When you talk about unsupervised neural networks it isn't clear how the "output mimics the input". For example supervised ML is more task driven problems, unsupervised is more data driven type problems. Can you give an example of how the output mimics the input.

6. line 68 - "We elaborate:" isn't common phrasing.

7. lines 70-73 - A figure or two would make this much easier to understand. If the goal of the paper is to help the clinician understand and appreciate (and believe) what ML can do for clinical medicine they will not understand this or likely take the time to try and understand it.

8. line 87 what is the "crossed" anterior ligament. Please, provide a table with the exact number of patients getting specific procedures. Because not all of these surgeries are equally as painful and pain cocktails used in spine surgery will inherently differ from those used in TJA. So this is potentially confounding.

9. was this data collected retrospectively or prospectively or a retrospective analysis of a prospective database?

10. Any study of pain medicine and the efficacy should include any preoperative narcotic or controlled substance use as this can have major effects on postoperative pain medicine utilization. Was there any attempt to quantify or control this?

11. Line 102 its about 50/50 general to regional anesthesia. Its important to note if the is neuroaxial (spinal/epidural), or peripheral nerve blocks like femoral nerve or saphenous nerve or iPACK for TKA. This is critical information along with the drugs that were used in the blocks. They all have different onset and duration times which will affect the utilization and efficacy of the scheduled pain meds and will differ between surgeries, surgeons, and anesthesiologists.

12. line 105 - please clarify points "a" what is "maximum flexing" - this sounds painful.

13. Were NRS only collected until 2pm on POD 1?

14. Table 1 - what is HDI 95%?.. Why is there so much descriptive detail about the patients' ages. Age is but just one variable in an orthopaedic study.

15. Line 195 - this just drops off, its an incomplete sentence.

16. line 235 -. Generally summarize what you found in 1 paragraph, 2nd paragraph how is it the same with what is in the literature, 3rd paragraph how is it different than what is in the literature, 4th - what other interesting variables that you found that have never been looked at before, 5th limitations, 6th conclusion.

17. Please, clarify the purpose and the implications of the autoencoder part of the analysis. Does the resultant two-dimensional representation of the input data contain any generalized knowledge or does it just directly encode the input? The encoding accuracy, e.g. the measured by the variance explained, is also not reported. It is therefore not unlikely that the clusters encode the most abundant combinations of analgesics in cocktails whereas the X and Y coordinates in the clusters encode the pre-and post-analgesia pain levels, which may be correlated. This can be tested via encoding the analgesics cocktail components without corresponding pain levels and seeing whether the five clusters are formed again. Third, there seem to be no conclusions derived from the obtained clusters. What would belonging to a certain cluster mean except for a high overlap in the list of the cocktail’s components? One way to use this embedding is to predict the pain reduction efficacy and significance for the cocktails not involved in the study via training another network on the two-dimensional embedding of the inputs. Finally, no argument is presented as to why autoencoder was used instead of simpler yet more interpretable alternatives such as CMDS, MDS, or Isomap.

Minor comments:

Please change “ML statistics” to “statistics”, and “AI” to “ML”.

Explain the choice of parameters for DBSCAN. For example, it seems reasonable to split cluster #2 into two clusters and to merge clusters #5 and #6 into one cluster.

Captions for Fig.1 and Fig. 2 appear swapped.

Contrary to the claim in the paper, cocktail choices are not a random (as in: i.i.d.) variable.

Please explain why the abundance of common ingredients in the same clusters deems surprising.

A marked-up copy of your manuscript that highlights changes made to the original version. You should upload this as a separate file labeled 'Revised Manuscript with Track Changes'.An unmarked version of your revised paper without tracked changes. You should upload this as a separate file labeled 'Manuscript'.

We look forward to receiving your revised manuscript.

Kind regards,

Gennady S. Cymbalyuk, Ph.D.

Academic Editor

PLOS ONE

Journal Requirements:

3. You indicated that you had ethical approval for your study. In your Methods section, please ensure you have also stated whether you obtained consent from parents or guardians of the minors included in the study or whether the research ethics committee or IRB specifically waived the need for their consent.

[The authors received financial support by the AUVA (Austrian working injury insurance company)]. 

Please respond by return email with your amended Competing Interests Statement and we will change the online submission form on your behalf.

7. Thank you for stating the following in the Acknowledgments Section of your manuscript: 

[We thank Ernst Reitbichler and the anesthesia nursing-team at the TZW Lorenz Boehler for their contribution in collecting data and Christopher Lockie, MD for language editing. The authors declare they have no conflicts of interest.

This study was enabled by financial support of the AUVA medical head office.]

 [GF recieved financial support by the AUVA (Austrian general working injury insurance company); the AUVA was not involved in any step of the generation of this scientific work;]

Reviewers' comments:

Reviewer's Responses to Questions

**Comments to the Author**

1. Is the manuscript technically sound, and do the data support the conclusions?

Reviewer #1: Partly

Reviewer #2: Yes

2. Has the statistical analysis been performed appropriately and rigorously? 

Reviewer #1: I Don't Know

Reviewer #2: Yes

3. Have the authors made all data underlying the findings in their manuscript fully available?

Reviewer #1: Yes

Reviewer #2: No

4. Is the manuscript presented in an intelligible fashion and written in standard English?

Reviewer #1: Yes

Reviewer #2: Yes

5. Review Comments to the Author

Reviewer #1: I congratulate the authors on what appears to be a technically heavy use of ML in an orthopaedic patient population. I am primarily a clinician (orthopedist) who has written about AI/ML but am by no means an expert. Any critiques I make regarding the TECHNOLOGY aspects of AI/ML are coming from someone with an intermediate fund of knowledge with the novice reader in mind.

I agree with the authors final statement that "For the patient, on the other hand, the ML

probability and its CI is of marginal importance". In other words, I don't know how this will change clinical practice - this is not a clinically impactful paper. But that may not be the authors' intent and shouldn't necessarily take away from what they did. This paper certainly had a feel of "here's how you COULD use ML to try and solve a clinical problem" as if it were written by somebody getting an advanced degree in AI/ML and this is their capstone project. And if that was the authors intention then I think there may be strength to publishing this paper. Because it is difficult for me to really ascertain the clinical impact of what was done here based on the clinical weaknesses. This very well could be a groundbreaking approach to analyzing a complex medical/pharmaceutical problem with ML but I have no way of knowing that.

There are many clinical variables to consider and are unanswered in the methods section. 1. many of these drugs I am not familiar with as they are not available in the US where I practice so I am uncertain of their efficacy. I recently sat on the American Association of Orthopaedic Surgeons CPG for "Pharmacologic, Physical, and Cognitive Pain Alleviation for Musculoskeletal Extremity/Pelvis Surgery" available at https://www.aaos.org/globalassets/quality-and-practice-resources/dod/painalleviationcpg.pdf. I had to look them up the drugs the authors listed that are unavailable in the US include Metamizole (NSAID), Dexibuprofen (NSAID), Piritramide (opioid). Furthermore it would appear that some patients received more than one type of opioid and/or NSAID - it is not common for patients in the US to receive more than one dose of 2 different types of NSAIDs - they may get celecoxib preoperatively and toradol postoperatively in the first 24 hours but I can't tell what was done here. 2. NSAIDs are the backbone of any multimodal pain pathway for orthopaedic surgery. I'm not aware that any one type of NSAID is superior to another NSAID. Infact in the AAOS CPG only Cox-2 inhibitors had enough high quality evidence to even qualify for inclusion in our review. 3. Probably most importantly the primary outcome (pain NRS) is becoming less emphasized in the US. We are fast tracking many orthopaedic surgeries (especially TJA) to discharge in less than 24 hours. I can't remember the last time I asked a patient to put a pain number on the knee or hip that I just effectively split or stretched tendons, sawed the bone, and jammed metal into. Our attitudes regarding pain have changed from "the fifth vital sign" with the recent US opioid epidemic. I think we have a pain control (pain expectation) problem in the US and shouldn't be emphasizing pain scores. MUCH more useful metrics include morphine equivalent dosing (MED), time to ambulation, distance ambulation, time to discharge, rescue pain medication use, etc. and must be taken at time points beyond the first 23 hours. Additionally a pain regimes ability to reduce NRS is only as important as its ability to avoid nausea/vomiting (opioid side effect), GI complications beyond N/V (gastric bleeding), decreased renal function (NSAID side effect), ability to use NSAIDs in cardiac, GI, renal patients, sedation, delirium, etc. Sedated and over-narcotized patients frequently will report less pain right before they have a hypoxic event.

I think the authors really need to re-frame this paper with the extreme clinical limitations in mind - this is not a clinical paper. I would frame this paper in the introduction as "here's how you could solve a complex medical challenge with ML", because it doesn't answer the question. But you would need to give a better explanation as THAT being the reason for performing this study and be honest up front that in no way is ML going to answer the question of "which of the 61 pain cocktails is most efficacious in 750 variable orthopaedic patients". And its possible that's what you did do but it was a technically difficult paper to read and not for the clinical orthopedist and you need to contextualize the clinical limitations of this paper.

Specific comments:

1. line 21 consider "potential" instead of "claimed" - efficacious synergy has been established.

2. line 46 - I would be specific about certain aspects of orthopaedic surgery - particularly the interest in "outpatient" total joints in the US, extremity trauma, and shoulder surgery (e.g. rotator cuff repair) are the most painful orthopedic surgeries. This is where pain control is most needed.

3. line 50 - I'm thinking in very clinically practical terms. I feel like this sentence requires a little more explanation. In the US we are frequently using toradol, oxycontin, NSAIDS, tylenol, etc. all together and we know that using them together is better than using any one of these drugs. What specific efficacy are you speaking about? Are you saying to what specific pain score or patient related outcome measure?

4. line 57 - I think it would be useful to in 1 or 2 sentences explain why conventional logistic regression cannot answer this question. Those unfamiliar with ML will not understand this.

5. line 59 - the first part of this sentence is confusing. Specifically the part "to learning algorithms labelled AI (artificial intelligence); the latter". I think you could get rid of that and the sentence is less confusing to read for the novice. ML really comes in 3 types - supervised, unsupervised, and reinforcement learning. In general lines 59-67 don't add much to this introduction. When you talk about unsupervised neural networks it isn't clear how the "output mimics the input". For example supervised ML is more task driven problems, unsupervised is more data driven type problems. Can you give an example of how the output mimics the input.

6. line 68 - "We elaborate:" isn't common phrasing.

7. lines 70-73 - I think a figure or two would make this much easier to understand. If the goal of the paper is to help the clinician understand and appreciate (and believe) what ML can do for clinical medicine they will not understand this or likely take the time to try and understand it. This is a major hurdle we need to overcome in order to bring AI to clinical medicine.

8. line 87 what is the "crossed" anterior ligament. There are 2 bundles in the ACL and 2 bundles in the PCL. The ACL and PCL together are known as the cruciate ligaments. While it is common to have an ACL reconstruction a ACL and PCL reconstruction in the same surgery (outside of a traumatic multi-ligament/knee dislocation) is very rare. Also what defines "complex reconstructive procedures". Are these revision TKAs and THAs or some sort of crushed extremity or other trauma. I think what will be helpful here is a table with the exact number of patients getting whatever procedures. For example TKA n = 68, ACL n = 45, ACL & MCL = 13, spine fusion (note if it was cervical or lumbar at a minimum) n = 45. Because not all of these surgeries are equally as painful and pain cocktails used in spine surgery will inherently differ from those used in TJA. So this is potentially confounding. I don't know how the vast number of cocktails (61) in a relatively small number of patients (750) is going to generate enough sampling for the ML.

9. was this data collected retrospectively or prospectively or a retrospective analysis of a prospective database?

10. Any study of pain medicine and the efficacy should include any preoperative narcotic or controlled substance use as this can have major effects on postoperative pain medicine utilization. Was there any attempt to quantify or control this?

11. Line 102 its about 50/50 general to regional anesthesia. Its important to note if the is neuroaxial (spinal/epidural), or peripheral nerve blocks like femoral nerve or saphenous nerve or iPACK for TKA. This is critical information along with the drugs that were used in the blocks. They all have different onset and duration times which will affect the utilization and efficacy of the scheduled pain meds and will differ between surgeries, surgeons, and anesthesiologists.

12. line 105 - please clarify points "a" what is "maximum flexing" - this sounds painful.

13. Were NRS only collected until 2pm on POD 1?

14. Table 1 - what is HDI 95%?..its clear I don't understand what is going on but why is there so much descriptive detail about the patients' ages. Age is but just one variable in an orthopaedic study.

15. Line 195 - this just drops off, its an incomplete sentence.

16. line 235 - I wouldn't open up the discussion with a question. Generally summarize what you found in 1 paragraph, 2nd paragraph how is it the same with what is in the literature, 3rd paragraph how is it different than what is in the literature, 4th - what other interesting variables that you found that have never been looked at before, 5th limitations, 6th conclusion.

Reviewer #2: In the paper, the authors investigate the highly important question of the efficacy of analgesics cocktails. Whereas their data is of high quality and the statistics on it are correct, the other part of the work which involves the low-dimensional embedding of the data remains inconclusive. I, therefore, recommend a revision. Please find the details below.

In this work, the authors predict the efficacy of various analgesic cocktails in reducing pain levels. To this end, the authors have collected a large-scale dataset of the patients’ reported pain levels before and after the use of analgesia. They used an autoencoder to obtain a two-dimensional embedding of the “cocktail components – pain levels” data points and used DBSCAN to cluster this embedding. The authors then have identified the most efficient cocktails via computing the statistics of whether pain levels were significantly decreased.

The goal of this work is highly important, and so is the collected dataset, as the data and analysis performed here help identify the more efficient analgesia.

The statistical analysis, to my understanding, is correct. The strength of the work is that the authors were able to identify analgesia with the most pronounced and statistically significant effect on reducing pain levels. These results can be readily used in clinical practice.

I am a bit less convinced of the purpose and the implications of the autoencoder part of the analysis. First – unless I am missing something here – it is unrelated to the rest of the analysis. Second, there is no argument presented as to whether the resultant two-dimensional representation of the input data contains any generalized knowledge or just directly encodes the input. The encoding accuracy, e.g. the measured by the variance explained, is also not reported. It is therefore not unlikely that the clusters encode the most abundant combinations of analgesics in cocktails whereas the X and Y coordinates in the clusters encode the pre-and post-analgesia pain levels, which may be correlated. This can be tested via encoding the analgesics cocktail components without corresponding pain levels and seeing whether the five clusters are formed again. Third, there seem to be no conclusions derived from the obtained clusters. What would belonging to a certain cluster mean except for a high overlap in the list of the cocktail’s components? One way to use this embedding is to predict the pain reduction efficacy and significance for the cocktails not involved in the study via training another network on the two-dimensional embedding of the inputs. Finally, no argument is presented as to why autoencoder was used instead of simpler yet more interpretable alternatives such as CMDS, MDS, or Isomap.

Minor comments:

Please change “ML statistics” to “statistics”, and “AI” to “ML”.

Explain the choice of parameters for DBSCAN. For example, it seems reasonable to split cluster #2 into two clusters and to merge clusters #5 and #6 into one cluster.

Captions for Fig.1 and Fig. 2 appear swapped.

Contrary to the claim in the paper, cocktail choices are not a random (as in: i.i.d.) variable.

Please explain why the abundance of common ingredients in the same clusters deems surprising.

Overall, I found the study interesting and relevant. The dataset on the effects of analgesics cocktails and the statistics on this dataset is ready for publication. The embedding part needs more clarity to be conclusive – along the lines mentioned above. I, therefore, recommend a revision.

6. PLOS authors have the option to publish the peer review history of their article (what does this mean?). If published, this will include your full peer review and any attached files.

Reviewer #1: **Yes: **Thomas Myers, MD, MPT

Reviewer #2: No

---

## [Author Response · Author response to Decision Letter 0]

20 Jun 2022

thank you for your effort to improve our work. Please find our comments in the specific file uploaded

---

## [Decision Letter · Decision Letter 1]

19 Jul 2022

PONE-D-21-40117R1Artificial intelligence algorithms predict the efficacy of analgesic cocktails prescribed after orthopedic surgeryPLOS ONE

Dear Dr. Fritsch,

Thank you for submitting your manuscript to PLOS ONE. After careful consideration, we feel that it has merit but does not fully meet PLOS ONE’s publication criteria as it currently stands. Therefore, we invite you to submit a revised version of the manuscript that addresses the points raised during the review process.

Please, edit the discussion to emphasize the technical side of the study and mitigate the strong conclusions claiming this manuscript to be a clinically relevant study. Please, acknowledge the limitations listed by the reviewers. Please, edit the statements in lines 350-352. Please, provide your responses with specific line number additions and bolded text so the changes could be easily seen in the manuscript. 

We look forward to receiving your revised manuscript.

Kind regards,

Gennady S. Cymbalyuk, Ph.D.

Academic Editor

PLOS ONE

Journal Requirements:

Reviewers' comments:

Reviewer's Responses to Questions

**Comments to the Author**

1. If the authors have adequately addressed your comments raised in a previous round of review and you feel that this manuscript is now acceptable for publication, you may indicate that here to bypass the “Comments to the Author” section, enter your conflict of interest statement in the “Confidential to Editor” section, and submit your "Accept" recommendation.

Reviewer #1: All comments have been addressed

Reviewer #2: (No Response)

2. Is the manuscript technically sound, and do the data support the conclusions?

Reviewer #1: Partly

Reviewer #2: Yes

3. Has the statistical analysis been performed appropriately and rigorously? 

Reviewer #1: I Don't Know

Reviewer #2: Yes

4. Have the authors made all data underlying the findings in their manuscript fully available?

Reviewer #1: Yes

Reviewer #2: Yes

5. Is the manuscript presented in an intelligible fashion and written in standard English?

Reviewer #1: Yes

Reviewer #2: Yes

6. Review Comments to the Author

Reviewer #1: I appreciate the effort the authors undertook to produce this manuscript. Its is difficult to tell if my previous comments were incorporated as there isn't a line by line response that I was able to find. I am going to suggest this be required in the future for any reviews to make this process easier for both parties.

I feel that this is a a technically sophisticated study and worth publishing based on the pretense that this is what someone *may* use AI/ML for in orthopaedics. I think it has major inherent weaknesses due to the breath of included orthopaedic surgeries each with their own type of patient population and I would caution the authors in making too strong of conclusions. For example a clinically relevant study would be looking at the combination of pain cocktails and nerve blocks (femoral, saphenous, iPACK, etc.) around only a TKA population. Including pediatric "knee surgery", athletic ACL, TKA, knee fractures is a very different type of study with many different surgeries. We would also need to knw if patients received general vs. neuroaxial blocks (e.g. spinals and how good is the clinician providing the spinal). Furthermore, there was no ability for the authors to control for a number of patient level confounding factors such as previous narcotic use, previous drugs of abuse, fear avoidance behaviors (kinesiophobia s/p surgery), depression/anxiety, fibromylagia, etc. Finally, you have to base your outcomes on more than numerical pain scores. There needs to be some quantification of patient reported outcomes, morphine equivalent doses of rescue med taken, etc.

Therefore, I would frame the discussion in this light and you must acknowledge these limitations. AI/ML is not going to compensate for a strong study design accounting for the the issues that I've previously mentioned - I would strongly disagree with the statements in lines 350-352 if that is what the authors are implying.

Any future edits need to incorporate these suggestions with specific line number additions and bolded text so I can see what has changed in the manuscript. Thanks you for your efforts!

Reviewer #2: (No Response)

7. PLOS authors have the option to publish the peer review history of their article (what does this mean?). If published, this will include your full peer review and any attached files.

Reviewer #1: **Yes: **Thomas Myers, MD, MPT

Reviewer #2: No

---

## [Author Response · Author response to Decision Letter 1]

4 Sep 2022

the answers on the comments of the reviewers are attached in the rebuttal letter attached to the manuscript

---

## [Decision Letter · Decision Letter 2]

20 Sep 2022

PONE-D-21-40117R2Artificial intelligence algorithms predict the efficacy of analgesic cocktails prescribed after orthopedic surgeryPLOS ONE

Dear Dr. Fritsch,

Thank you for submitting your manuscript to PLOS ONE. After careful consideration, we feel that it has merit but does not fully meet PLOS ONE’s publication criteria as it currently stands. Therefore, we invite you to submit a revised version of the manuscript that addresses the points raised during the review process. Could you address the concerns of the reviewer by accepting that the list of factors requested by the reviewer are important and could be included in future applications?

We look forward to receiving your revised manuscript.

Kind regards,

Gennady S. Cymbalyuk, Ph.D.

Academic Editor

PLOS ONE

Journal Requirements:

Reviewers' comments:

Reviewer's Responses to Questions

**Comments to the Author**

1. If the authors have adequately addressed your comments raised in a previous round of review and you feel that this manuscript is now acceptable for publication, you may indicate that here to bypass the “Comments to the Author” section, enter your conflict of interest statement in the “Confidential to Editor” section, and submit your "Accept" recommendation.

Reviewer #1: All comments have been addressed

Reviewer #2: All comments have been addressed

2. Is the manuscript technically sound, and do the data support the conclusions?

Reviewer #1: No

Reviewer #2: Yes

3. Has the statistical analysis been performed appropriately and rigorously? 

Reviewer #1: I Don't Know

Reviewer #2: N/A

4. Have the authors made all data underlying the findings in their manuscript fully available?

Reviewer #1: Yes

Reviewer #2: No

5. Is the manuscript presented in an intelligible fashion and written in standard English?

Reviewer #1: Yes

Reviewer #2: Yes

6. Review Comments to the Author

Reviewer #1: I read the manuscript starting on page 45 because it was label "revised manuscript with track changes". That being said the only text with red font was in 351 - 356, and another section after this.

I haven't changed my thoughts on this paper since my last set of comments. Overall I feel that this paper speaks to the techniques of machine learning as much as it does sound clinical science. I think this paper's strength and publication worthiness is based on the fact that this is what someone *may* use machine learning for in orthopaedics/anesthesia. It DOES NOT show which pain medications are best to use for any specific (subspecialty) orthopaedic surgery.

I don't think this study answers the clinical questions of what pain regimens are best after orthopedic surgery. I have copy and pasted my previous comment below for the editors (in quotations). Thank you for your efforts.

"I feel that this is a a technically sophisticated study and worth publishing based on the pretense that this is what someone *may* use AI/ML for in orthopaedics. I think it has major inherent weaknesses due to the breath of included orthopaedic surgeries each with their own type of patient population and I would caution the authors in making too strong of conclusions. For example a clinically relevant study would be looking at the combination of pain cocktails and nerve blocks (femoral, saphenous, iPACK, etc.) around only a TKA population. Including pediatric "knee surgery", athletic ACL, TKA, knee fractures is a very different type of study with many different surgeries. We would also need to know if patients received general vs. neuroaxial blocks (e.g. spinals and how good is the clinician providing the spinal). Furthermore, there was no ability for the authors to control for a number of patient level confounding factors such as previous narcotic use, previous drugs of abuse, fear avoidance behaviors (kinesiophobia s/p surgery), depression/anxiety, fibromylagia, etc. Finally, you have to base your outcomes on more than numerical pain scores. There needs to be some quantification of patient reported outcomes, morphine equivalent doses of rescue med taken, etc."

Reviewer #2: (No Response)

7. PLOS authors have the option to publish the peer review history of their article (what does this mean?). If published, this will include your full peer review and any attached files.

Reviewer #1: No

Reviewer #2: No

---

## [Author Response · Author response to Decision Letter 2]

30 Nov 2022

Please find our comments within the rebuttal letter attached

---

## [Decision Letter · Decision Letter 3]

13 Jan 2023

Artificial intelligence algorithms predict the efficacy of analgesic cocktails prescribed after orthopedic surgery

PONE-D-21-40117R3

Dear Dr. Fritsch,

We’re pleased to inform you that your manuscript has been judged scientifically suitable for publication and will be formally accepted for publication once it meets all outstanding technical requirements.

Kind regards,

Gennady S. Cymbalyuk, Ph.D.

Academic Editor

PLOS ONE

Additional Editor Comments (optional):

Reviewers' comments:

Reviewer's Responses to Questions

**Comments to the Author**

1. If the authors have adequately addressed your comments raised in a previous round of review and you feel that this manuscript is now acceptable for publication, you may indicate that here to bypass the “Comments to the Author” section, enter your conflict of interest statement in the “Confidential to Editor” section, and submit your "Accept" recommendation.

Reviewer #1: All comments have been addressed

2. Is the manuscript technically sound, and do the data support the conclusions?

Reviewer #1: Partly

3. Has the statistical analysis been performed appropriately and rigorously? 

Reviewer #1: I Don't Know

4. Have the authors made all data underlying the findings in their manuscript fully available?

Reviewer #1: Yes

5. Is the manuscript presented in an intelligible fashion and written in standard English?

Reviewer #1: Yes

6. Review Comments to the Author

Reviewer #1: nothing further to add i would refer to my previous comments. I would leave the publication of this paper up to the editors.

7. PLOS authors have the option to publish the peer review history of their article (what does this mean?). If published, this will include your full peer review and any attached files.

Reviewer #1: **Yes: **Thomas Myers

---

## [Editor Report · Acceptance letter]

20 Jan 2023

PONE-D-21-40117R3 

Artificial intelligence algorithms predict the efficacy of analgesic cocktails prescribed after orthopedic surgery 

Dear Dr. Fritsch:

I'm pleased to inform you that your manuscript has been deemed suitable for publication in PLOS ONE. Congratulations! Your manuscript is now with our production department. 

Kind regards, 

on behalf of

Dr. Gennady S. Cymbalyuk 

Academic Editor

PLOS ONE